# Local hypergraph clustering using capacity releasing diffusion

**Rania Ibrahim●\*, David F. Gleich**

Computer Science Department, Purdue University, West Lafayette, Indiana, United States of America

\* ibrahimr@purdue.edu

## Abstract

Local graph clustering is an important machine learning task that aims to find a well-connected cluster near a set of seed nodes. Recent results have revealed that incorporating higher order information significantly enhances the results of graph clustering techniques. The majority of existing research in this area focuses on spectral graph theory-based techniques. However, an alternative perspective on local graph clustering arises from using max-flow and min-cut on the objectives, which offer distinctly different guarantees. For instance, a new method called capacity releasing diffusion (CRD) was recently proposed and shown to preserve local structure around the seeds better than spectral methods. The method was also the first local clustering technique that is not subject to the quadratic Cheeger inequality by assuming a good cluster near the seed nodes. In this paper, we propose a local hypergraph clustering technique called hypergraph CRD (HG-CRD) by extending the CRD process to cluster based on higher order patterns, encoded as hyperedges of a hypergraph. Moreover, we theoretically show that HG-CRD gives results about a quantity called motif conductance, rather than a biased version used in previous experiments. Experimental results on synthetic datasets and real world graphs show that HG-CRD enhances the clustering quality.

## 1 Introduction

Graph and network mining techniques traditionally experience a variety of issues as they scale to larger data [1]. For instance, methods can take prohibitive amounts of time or memory (or both), or simply return results that are trivial. One important class of methods that has a different set of trade-offs are local clustering algorithms [2]. These methods seek to apply a graph mining, or clustering (in this case), procedure around a seed set of nodes, where we are only interesting in the output nearby the seeds. In this way, local clustering algorithms avoid the memory and time bottlenecks that other algorithms experience. They also tend to produce more useful results as the presence of the seed is powerful guidance about what is relevant (or not). For more about the trade-offs of local clustering and local graph analysis, we defer to the surveys [3].

Among the local clustering techniques, the two predominant paradigms [3] are (i) spectral algorithms [2, 4–6], that use random walk, PageRank, and Laplacian methodologies to identify

**Data Availability Statement:** The following datasets were used in this study: Stanford Large Network Dataset Collection: https://snap.stanford.edu/data/. Large hypergraph experiments: https://www.cs.cornell.edu/~arb/data/.

**Funding:** This work is supported by NSF awards IIS-1546488, CCF-1909528 NSF Center for Science of Information STC, CCF-0939370, DE-SC0014543, NASA, and the Sloan Foundation.

**Competing interests:** The authors have declared that no competing interests exist.

good clusters nearby the seed and (ii) mincut and flow algorithms [7–11], that use parametric linear programs as well as max-flow, min-cut methodologies to identify good clusters nearby the seed. Both have different types of trade-offs. Mincut-based techniques often better optimize objectives such as *conductance* or *sparsity* whereas spectral techniques are usually faster, but slightly less precise in their answers [10, 12]. These differences often manifest in the types of theoretical guarantees they provide, usually called *Cheeger inequalities* for spectral algorithms. A recent innovation in this space of algorithms is the capacity releasing diffusion, which can be thought of as a hybrid max-flow, spectral algorithm in that it combines some features of both. This procedure provides excellent recovery of many node attributes in labeled graph experiments in Facebook, for example.

In a different line of work on graph mining, the importance of using higher-order information embedded within a graph or data has recently been highlighted in a number of venues [13–15]. In the context of graph mining, this usually takes the form of building a hypergraph from the original graph based on a motif [15]. Here, a motif is just a small pattern, think of a triangle in a social network, or a directed pattern in other networks like a cycle or feed-forward loop patterns. The hypergraph corresponds to all instances of the motif in the network. Analyzing these hypergraphs usually gives much more refined insight into the graph. This type of analysis can be combined with local spectral methods too, as in the MAPPR method [16]. However, the guarantees of the spectral techniques are usually biased for large motifs because they implicitly, or explicitly, operate on a clique expansion representation of the hypergraph [15].

In this paper, we present HG-CRD, a hypergraph-based implementation of the capacity releasing diffusion hybrid algorithm that combines spectral-like diffusion with flow-like guarantees. In particular, we show that our method provides cluster recovery guarantees in terms of the true motif conductance, not the approximate motif conductance implicit in many spectral hypergraph algorithms [13, 16]. The key insight to the method is establishing an algorithm that manages flow over hyperedges induced by the motifs. More precisely, if we use for illustration, a triangle as our desired motif, if a node $i$ wants to send a flow to node $j$ in the process, instead of sending the flow through the edge $(i, j)$, it will send the flow through the hyperedge $(i, j, k)$. This ensures that node $i$ sends flow to node $j$ that is connected to it via a motif and that nodes $i$, $j$ and $k$ are explored simultaneously. To show why is it important to consider higher order relations, we explore a similar metabolic network explored by Li and Milenkovic [15], where nodes represent metabolites and edges represent the interactions between metabolites. These interactions usually described by equations as $M1 + M2 \rightarrow M3$, where M1 and M2 are the reactant and M3 is the product of the interaction. Our goal here is to group metabolite represented by node 7 with other nodes based on their metabolic interactions. Fig 1 shows the used motif in HG-CRD and the graph to cluster. By considering this motif, HG-CRD separates three metabolic interactions, while CRD separates six metabolic interactions. Note that, for three node motifs, Benson et al. [13] show that a weighted-graph called the motif adjacency matrix, suffices for many applications. We also consider this weighted approach via an algorithm called CRD-M (here, M, is for motif matrix), and find that the HG-CRD approach is typically, but not always, superior.

Additionally, we show that HG-CRD gives a guarantee in terms of the exact motif-conductance score that mirrors the guarantees of the CRD algorithm. More precisely, if there is a good cluster nearby, defined in terms of motif-conductance, and that cluster is well connected internally, then the HG-CRD algorithm will find it. This is formalized in section 4.6.

Finally, experimental results on both synthetic datasets and real world graphs and hypergraphs for community detection show that HG-CRD has a lower motif conductance than the

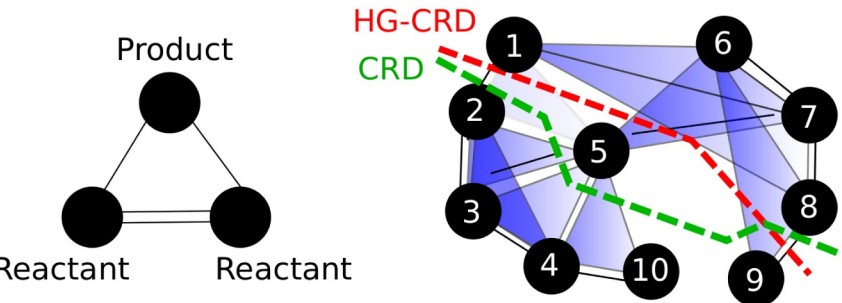

**Fig 1. Running CRD and HG-CRD on the metabolic graph [15], shows that CRD returns a set with motif conductance of 0.33 while HG-CRD has motif conductance of 0.27.** (Algorithm parameters for reproducibility: $h = 2$, $C = 2$, $\tau = 2$, iterations = 5, $\alpha = 1$ and starting from node 7.). The motif hypergraph has edges: {1,2,5}, {2,3,5}, {3,4,5}, {4,5,10}, {2,3,4}, {1,6,7}, {1,7,8}, {5,6,7} and {6,8,9}.

original CRD in most of the datasets and always has a better precision in all datasets. To summarize our contributions:

- We propose a local hypergraph clustering technique HG-CRD by extending the capacity releasing diffusion process to account for hyperedges in section 4.

- We show that HG-CRD is the first local higher order graph clustering method that is not subject to the quadratic Cheeger inequality in section 4.6 by assuming the existence of a good cluster near the seed nodes.

- We compare HG-CRD to the original CRD and other related work on both synthetic datasets and real world graphs and hypergraphs for community detection in section 5. These experiments show HG-CRD offers new opportunities for reliable hypergraph-based community detection in a variety of scenarios.

## 2 Related work

While there is a tremendous amount of research on local graph mining, we focus our attention on the most closely related ideas that have influenced our ideas in this manuscript. This includes hypergraph clustering and higher-order graph analysis, local clustering, the capacity releasing diffusion method, and the MAPPR method.

### 2.1 Hypergraph clustering and higher-order graph analysis

It is essential to develop graph analysis techniques that exploit higher order structures, as shown by [17, 18], because these higher order structures reveal important latent organization characteristics of graphs that are hard to reveal from the edges alone. Several techniques [13–15] are available for the general analysis of higher-order clustering or hypergraph clustering. The two are related because the higher-order structures, or motifs, as used in [13], can be assembled into a hypergraph representation where each hyperedge expresses the presence of a motif. In all three methods, the hypergraph information is re-encoded into a weighted graph or weighted adjacency matrix. For instance, in [13], they construct an $n \times n$ motif adjacency matrix, where each entry $(i, j)$ in the matrix represents the number of motif instances that contain both node $i$ and node $j$. Then, it uses the motif adjacency matrix as an input to the spectral clustering technique. This has good theoretical guarantees only for 3 node motifs. Likely, [14] also constructs the same type of motif adjacency matrix. Using the motif adjacency matrix, Tectonic normalizes the edge weights of the motif adjacency matrix by dividing over the

degree of the motif nodes. Finally, to detect the clusters, Tectonic removes all edges with weight less than a threshold $\theta$. More recently, [15] generalize the previous results by assigning different costs for different partitions of a hyperedge before reducing the hypergraph to a matrix. They also show that the returned cluster conductance has a quadratic approximation to the optimal conductance and give the first bound on the performance for a general sized hyperedge. See another recent paper [19] for additional discussion and ideas regarding hypergraph constructions and cuts.

*In contrast, in our work here, our goal is an algorithm that directly uses the hyperedges without the motif-adjacency matrix construction entirely, i.e. the HG-CRD method.* We use the motif-weighted adjacency matrix (CRD-M) solely for comparison.

For spectral-based hypergraph learning, references [20, 21] study a variety of hypergraph laplacians (e.g. those based on clique and star expansions and Rodriguez's formulations) and use those to generalize spectral clustering for hypergraphs. More recently, references [22, 23] use submodular function minimization for hypergraph learning, which are still spectral-type algorithms. None of the previous techniques use local clustering or algorithms as we do here. Our objective in this paper is to extend the CRD process for hypergraphs since it was shown to produce more localized clusters than a variety of spectral-based techniques.

## 2.2 Local clustering

As mentioned in the introduction, local clustering largely splits along the lines of spectral approaches—those that use random walks, PageRank, and Laplacian matrices—and mincut approaches—those that use linear programming and max-flow methodologies [3]. For instance, approximate Personalized PageRank approaches due to [2] are tremendously successful in revealing local structure in the graph nearby a seed set of vertices. Examples abound and include the references. [24–28]. We elaborate more on these below because we use a motif-weighted local clustering algorithm MAPPR as a key point of comparison with this class of techniques.

Mincut based techniques [7–9] form a sequence or a parametric linear program that is, under strong assumptions, capable of optimally solving for the *best* local set in terms of objectives such as set conductance (number of edges leaving divided by total number of edges) and set sparsity (number of edges leaving divided by number of vertices). These can be further localized [9–11] by incorporating additional objective terms to find smaller sets. As mentioned, a recent innovation is the CRD algorithm, which presents a new set of opportunities. We explain more about that algorithm below. Related ideas exist for metrics such as modularity [29].

## 2.3 MAPPR

Motif-based approximate personalized PageRank (MAPPR) [16] is a local higher order graph clustering technique that generalizes approximate personalized PageRank (APPR) [2] to cluster based on motifs instead of edges. MAPPR was proven to detect clusters with small motif conductance and with a running time that depends on the size of the cluster. Experimental results on community detection show that MAPPR outperforms APPR in the quality of the detected communities. Like the existing work, MAPPR uses the motif-weighted adjacency matrix. Again, our goal is to avoid this construction, although we use it for comparison.

## 2.4 Capacity releasing diffusion

[30] proposed a new diffusion technique called capacity releasing diffusion (CRD), which is faster and stays more localized than spectral diffusion methods. Additionally, CRD is the first

diffusion method that is not subject to the quadratic Cheeger inequality. The basic idea of the CRD process is to assume that each node has a certain capacity and then start with putting excess of flow on the seed nodes. After that, let the nodes transmit their excess of flow to other nodes according to a similar push/relabel strategy proposed by [31]. If all the nodes end up with no excess of flow, then there was no bottleneck that kept the flow contaminated and therefore CRD repeats the same process again while doubling the flow at the nodes that are visited. On the other hand, if at the end of the iteration, we observed that too many nodes have too much excess (according to some parameter), then we have hit a bottleneck and we should stop and return the cluster. The cluster will be identified by the nodes that have excess of flow at the end of the iteration. In this work, we extend CRD process to consider clustering based on higher order structures.

## 2.5 Key differences with our contribution

In this work, we aim to extend capacity releasing diffusion process (CRD) to account for clustering based on higher order structures instead of edges. As CRD was shown to stay more localized than spectral diffusions, higher order techniques based on CRD will also stay more localized than spectral-based higher order techniques like MAPPR, this is the main motivation of why we choose to extend the CRD process. Furthermore, we discuss this localization property in more details in section 4.5.

# 3 Local cluster quality

Two important measures to quantify a set $S$ as a cluster in the graph $G$ are conductance [32] and motif conductance [13]. We define these here, as well as reviewing our general notation.

Scalars are denoted by small letters (e.g., $m$, $n$), sets are shown in capital letters (e.g., $X$, $Y$), vectors are denoted by small bold letters (e.g., f, g), and matrices are capital bold (**A**, **B**). Given an undirected graph $G = (V, E)$, where $V$ is the set of nodes and $E$ is the set of edges, we use **d**$(i)$ to denote the degree of the vertex $i$, and we use $n$ to be the number of nodes in the graph. Additionally, let **e** be the vector of all ones of appropriate dimension.

**Conductance** captures both how well-connected set $S$ is internally and externally and it is defined in terms of the *volume* and *cut* of the set. The volume of set $S$ is vol$(S) = \Sigma_{v \in S}$ **d**$(v)$. The cut of set $S$ is the set of all edges where one end-point is in $S$ and the other is not. This gives $\text{cut}(S) = \{\{u, v\} \in E : u \in S \text{ and } v \in \bar{S}\}$. Then the conductance of the set $S$ is defined as:

$$\phi(S) = \frac{|\text{cut}(S)|}{\text{minvol}(S)},$$

where minvol$(S) = \min(\text{vol}(S), \text{vol}(V - S))$. For weighted graphs, the volume uses the weighted degrees and the $|\text{cut}(S)|$ is the sum of edge weights cut.

**Motif conductance** as defined by [13] is a generalization of the conductance to measure the clustering quality with respect to a specific motif instance, it is defined as:

$$\phi_M(S) = \frac{|\text{cut}_M(S)|}{\text{minvol}_M(S)},$$

where $\text{cut}_M(S)$ is the number of motif instances that have at least one node in $S$ and at least one node in $\bar{S}$, $\text{vol}_M(S)$ is the number of motif instance end points in $S$ and $\text{minvol}_M(S) = \min(\text{vol}_M(S), \text{vol}_M(V - S))$. Additionally, we will define the size of the motif $k$ as the number of nodes in the motif.

**The motif hypergraph** is built on the same node set as $G$ where each hyperedge represents an instance of a single motif. For instance, the motif hypergraph for Fig 1 has edges: {1,2,5},

{2,3,5}, {3,4,5}, {4,5,10}, {2,3,4}, {1,6,7}, {1,7,8}, {5,6,7} and {6,8,9}. The cut of a set of vertices *S* in a hypergraph is the set of hyperedges that have an end point in *S* and another end point outside of *S*. Thus, the motif cut $|\mathrm{cut}_M(S)|$ and the hypergraph cut in the motif hypergraph are identical. There are some subtleties in the definition of degree and volume, which we revisit shortly.

## 4 A Capacity releasing diffusion for hypergraphs via motif matrices

Our main contribution is the HG-CRD algorithm that avoids the motif-weighted adjacency matrix. We begin, however, by describing how CRD can already be combined with the motif-weighted adjacency matrix in the CRD-M algorithm. For three node motifs, this offers a variety of theoretical guarantees, but there are significant biases for larger motifs that we are able to mitigate with our HG-CRD algorithm.

One straightforward idea to extend capacity releasing diffusion to cluster based on higher order patterns is to construct the motif adjacency matrix $W_M$ as described by [13], where the entry $W_{M(i,\,j)}$ equals to the number of motif instances that both node *i* and node *j* appear in its nodes. After constructing the motif adjacency matrix, we will run the CRD process on the graph represented by the motif adjacency matrix $W_M$. As the CRD process does not take into account the weight of the edges, we consider two variations, the first one is to duplicate each edge $(u, v)$ $W_{M(u,\,v)}$ times. This results in a multigraph, where the CRD algorithm can be easily adapted and the second variation is to multiply the weight of the edge $(u, v)$ used in the CRD process by $W_{M(u,\,v)}$. We report the second variation in the experimental results as it has the best $F_1$ in all datasets. Based on the proof presented in [13], when the size of the motif is three, then the motif conductance will be equal to the conductance of the graph represented by the motif adjacency matrix. As the CRD process guarantees that the conductance of the returned cluster is $O(\phi)$ where $\phi$ is a parameter input to the algorithm that controls both the maximum flow that can be pushed along each edge and the maximum node level during the push-relabel process. This extension guarantees that when the motif size is three, the motif conductance of the returned cluster is $O(\phi)$. In the rest of the paper, we will propose another extension for higher order capacity releasing diffusion that has a motif conductance guarantee of $O(k\phi)$ for any motif size *k*.

### 4.1 A true hypergraph CRD

Let $H = (V, \mathcal{E})$ be a hypergraph where the hyperedges $\mathcal{E}$ represent motifs. For this reason, we will use *motif* and *hyperedge* largely interchangeably. The basic idea in CRD is to push flow along edges until a bottleneck emerges. In HG-CRD, the basic idea is the same. We push the flow across hyperedges as much as we can until there is too much excess flow on the nodes, which means nodes are unable to push much flow to hyperedges containing them. This will happen because the algorithm has reached a bottleneck and hence the algorithm has identified the desired cluster with respect to the high order pattern or motif. As such, the HG-CRD procedure is highly algorithmic as it implements a specific procedure to move *flow* around the graph to identify this bottleneck.

The input to the HG-CRD algorithm is a hypergraph H, a seed node *s*, and parameters $\phi$, $\tau$, $t$, $\alpha$. The parameter *t* controls the number of iterations of the procedure, which is largely correlated with how large the final set should be. The parameter $\tau$ controls what is too much excess flow on the nodes and the parameter $\alpha$ controls whether we can push flow along the hyperedge or not. The value $\phi$ is the target value of motif conductance to obtain. That is, $\phi$ is an estimate of the bottleneck. Note that much of the theory is in terms of the value of $\phi$, but the algorithm

uses the quantity $C = 1/\phi$ in many places instead. The output of the algorithm is the set $A$ representing the cluster of node $s$.

We begin by providing a set of quantities the algorithm manipulates and a simple example.

### 4.2 Node and hyperedge variables in HG-CRD

Each node $v$ in the graph will have four values associated with it, which are: the degree of the node $\mathbf{d}_{M(v)}$, the flow at the node $\mathbf{m}_{M(v)}$, the excess flow at the node $\mathbf{ex}(v)$, and the node level $\mathbf{l}(v)$, where each node level starts having value one similar to max flow algorithms. In the hypergraph, we define the degree as follows:

$$\mathbf{d}_M(v) = (k-1)\eta(v)$$

That is, $\eta(v)$ is the number of hyperedges containing $v$ and degree $\mathbf{d}_{M(v)}$ of vertex $v$ will be $(k-1)\times$ the number of hyperedges containing $v$.

A node $v$ has excess flow if and only if $\mathbf{m}_{M(v) \geq} {}_{\mathbf{d}M}(v)$ and in this case we will call node $v$ to be *active*. Additionally, we will define excess of flow $\mathbf{ex}(v)$ at node $v$ to be $\max(\mathbf{m}_{M(v) - \mathbf{d}M}(v), 0)$ and therefore, we visualize each node $v$ to have a capacity of $\mathbf{d}_{M(v)}$ and any extra flow than this is excess. We will also restrict the node level to have a maximum value $h = \frac{3\log(\mathbf{e}^T\mathbf{m}_M)}{\phi}$. If $\mathbf{l}(v)$ reach $h$, then node $v$ cannot send or receive any flow.

Moreover, each hyperedge $e = (v, u_1, \ldots, u_{k-1})$ will have a capacity $C = \frac{1}{\phi}$, where $\phi$ is a parameter input to the algorithm and $e$ will have a flow value associated with it $\mathbf{m}_{M(e)}$, which represents how much flow is pushed along this hyperedge. To determine if we can push more flow through this hyperedge or not, we will have a residual capacity variable with each hyperedge and it will be defined as $r(e) = \min(\mathbf{l}(v), C) - \mathbf{m}_M(e)$. In the process, we can push flow through the hyperedge $e = (v, u_1, \ldots, u_{k-1})$ if and only if it is an eligible hyperedge, where an eligible hyperedge is defined as:

- $\mathbf{l}(v) > \mathbf{l}(u_i)$ for at least $\alpha$ $u_i \in e \setminus v$, where $\alpha \in [1, k-1]$. For example, if $\alpha = 1$, then we need at least one node in the hyperedge with level less than the level of node $v$ so that we can consider this hyperedge as an eligible hyperedge. Additionally, if $\alpha = 2$, then we need at least two nodes in the hyperedge with levels less than the level of node $v$.

- $r(e) > 0$.

- $ex(v) \geq k - 1$.

```
 1 HG-CRD Algorithm(G, s, φ, τ, t, α)
 2 m_M ← 0; m_M(s) ← d_M(s); A ← {}; φ* ← 1
 3 foreach j = 0, ..., t do
 4    m_M ← 2m_M;
 5    l ← HG-CRD-inner(G, m_M, φ, α);
 6    K_j ← Sweepcut(G, l)
 7    if φ_M(K_j) < φ* then A ← K_j; φ* ← φ_M(K_j)
 8    m_M ← min(m_M, d_M)
 9    if e^T m_M ≤ τ(2d_M(s)2^j) then Return A
 end
10 Return A
11 HG-CRD-inner(G, m_M, φ, α
12 l = 0; ex = 0; Q = {v|m_M(v) ≥ d_M(v) + (k - 1)};
13 h = 3log((e^T m_M))/φ
14 while Q is not empty do
15    v ← Q.get-lowest-level-node();
16    e = (v, u_1, ..., u_{k-1}) ← pick-eligible-hyperedge(v);
```

```
17   pushed-flow ← false
18   if e ≠ {} then
19     Ψ = min(⌊ex(v)/(k-1)⌋, r(e), 2d_M(u_i) - m_M(u_i) ∀u_i);
20     m_M(e) ← m_M(e) + Ψ
21     m_M(v) ← m_M(v) - (k - 1)Ψ;
22     m_M(u) ← m_M(u) + Ψ ∀u_i ∈ e \ v
23     if Ψ ≠ 0 then
24       pushed-flow ← true;
25       ex(v) ← max(m_M(v) - d_M(v), 0)
26       if ex(v) < k - 1 then Q.remove(v)
27       for u_i ∈ e do
28         ex(u_i) ← max(m_M(u_i) - d_M(u_i), 0)
29         if ex(u_i) ≥ (k - 1) and l(u_i) < h then
30         Q.add(u_i)
      end
   end
31   if not pushed-flow then
32     l(v) ← l(v) + 1 if l(v) == h then Q.remove(v)
   end
end
```

## 4.3 The high-level algorithm

The exact algorithm details which are adjusted from the pseudocodes in [30] to account for the impact of hyperedges, are present in HG-CRD Algorithm. In this section, we summarize the main intuition for the algorithm. Table 1 shows the description of used terms in the pseudo-code. At the start, the flow value $\mathbf{m}_{M(s)}$ at the seed node $s$ will be equal to $2\mathbf{d}_M(s)$ and in each iteration, HG-CRD will double the flow on each visited node. Each node $v$ that has excess flow picks an *eligible* hyperedge $e$ that contains the node $v$ and sends flow to all other nodes in the hyperedge $e$. After performing an iteration (which we will call it HG-CRD-inner in the pseudo-docode,) we will remove any excess flow at the nodes (step 8 in pseudocode). If all nodes do

**Table 1. Description of used terms.**

| Term | Definition |
| --- | --- |
| $\mathbf{d}_{M(v)}$ | motif-based degree of node $v$ |
| $\mathbf{m}_{M(v)}$ | flow at node $v$ |
| $\mathbf{ex}(v)$ | excess flow at node $v$ |
| $\mathbf{l}(v)$ | level of node $v$ |
| $h$ | maximum level of any node |
| $C$ | maximum capacity of any edge |
| $\mathbf{m}_{M(e)}$ | flow at edge $e$ |
| $r(e)$ | residual capacity of edge $e$ |
| $s$ | seed node |
| $t$ | maximum number of HG-CRD inner calls |
| $\tau$ | controls how much is much excess |
| $\alpha$ | controls the eligibility of hyperedge |
| $\phi$ | controls the values of $C$ and $h$ |
| $\mathbf{e}$ | vector of all ones |
| $\text{vol}_M(B)$ | the number of motif instances end points in $B$ |
| $\text{volM}(B)$ | $\Sigma_{v \in B} d_M(v) = (k - 1)\text{vol}_M(B)$ |
| $\phi_M(B)$ | motif conductance of $B$ |
| $K_j$ | set of nodes in the cluster in the jth iteration |

not have excess flow in all iterations (no bottleneck is reached yet) and as we double the flow at each iteration $j$, then the total sum of flow at the nodes will be equal to $2\mathbf{d}_M(s)2^j$. However, if the total sum of flow at the nodes is significantly (according to the parameter $\tau$) less than $2\mathbf{d}_M(s)2^j$, then step 8 has removed a lot of flow excess at the nodes because many nodes had flow excess at the end of the iteration. This indicates that the flow was contaminated and we have reached a bottleneck for the cluster. Finally, we will obtain the cluster $A$ by performing a sweep-cut procedure. Sweep-cut is done by sorting the nodes descendingly according to their level values, then evaluating the conductance of each prefix set and returning the cluster with the lowest conductance. This will obtain a cluster with motif conductance of $O(k\phi)$ as shown in section 4.6.

## 4.4 HG-CRD toy example

In this section, we will describe the hypergraph CRD dynamics on a toy example. Fig 2 shows the motif-based degree and the flow for each node of the graph at each iteration. We will start the process from the seed node 0 and let us choose the triangle as the motif we would like to consider in our clustering process. Additionally, we will fix each hyperedge maximum capacity $C$ to be two. At iteration 0, node 0 will increase its level by one in order to be able to push flow, then it will pick hyperedges {0, 1, 3}, {0, 1, 2} and {0, 2, 3} and push one unit of flow to each one of them. After that at iteration 1, each node will double its flow value. In this iteration, only node 0 has excess of flow and therefore it will again pick hyperedge {0, 1, 3}, {0, 1, 2} and {0, 2, 3} and push one unit of flow to each one of them. Similarly at iteration 2, each node will double its flow value. In this iteration, nodes 0, 1, 2 and 3 have excess of flow and therefore node 1 will send two units of flow to hyperedge {1, 4, 6} and will not be able to send more flow as the hyperedge has reached its maximum capacity (recall that we have fixed the maximum capacity of each hyperedge to be two). In the rest of the iteration, node 0, 1, 2 and 3 will exchange the flow between them until their levels reach the maximum level $h$ and this will terminate the iteration. Finally, we have four nodes with excess and the detected cluster $A$ will be 0, 1, 2 and 3 as these nodes have $\mathbf{m}_M \geq \mathbf{d}_M$. In case of original CRD, the returned cluster $A$ will be nodes 0, 1, 2, 3 and 7, as some flow will leak to node 7.

## 4.5 Motivating example

One of the key advantages of CRD and our HG-CRD generalization is that they can provably explore a vastly smaller region of the graph than random walk or spectral methods. Even those based on the approximate PageRank. Here, Fig 3 shows a generalized graph of the graph provided in the original CRD paper [30], where we are interested in clustering this graph based on the triangle motif. In this graph, there are $p$ paths, each having $l$ triangles and each path is connected at the end to the vertex $u$ with a triangle. In higher order spectral clustering techniques, the process will require $\Omega(k^2 l^2)$ steps to spread enough mass to cluster $B$. During these steps, the probability to visit node $v$ is $\Omega(kl/p)$. If $l = \Omega(p)$, then the random walk will escape from $B$. However, let us consider the worst case in HG-CRD where we start from $u$. In this case, $1/p$ of the flow will leak from $u$ to $v$ in each iteration. As HG-CRD process doubles the flow of the nodes in each iteration, it only needs $\log l$ iterations to spread enough flow to cluster $B$ and therefore the flow leaking to $\bar{B}$ will be $(\log l)/p$ of the total flow in the graph. This means HG-CRD will stay more localized in cluster $B$ and will leak flow to $\bar{B}$ much less than higher order spectral clustering techniques by a factor of $\Omega\left(\frac{p}{\log l}\right)$. HG-CRD is also able to detect the cluster in fewer number of iterations than higher order spectral clustering.

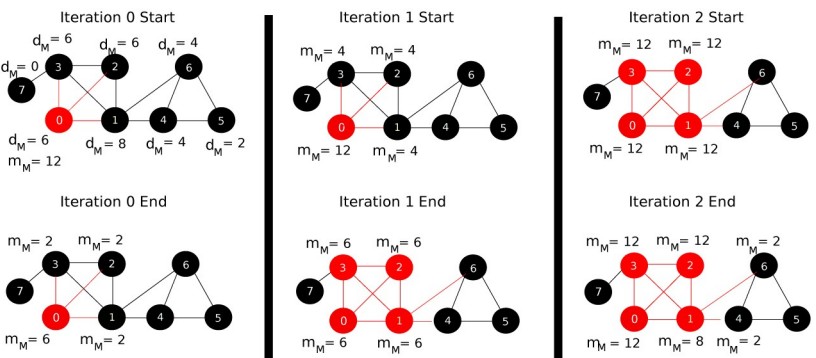

**Fig 2. Explanation of hypergraph CRD (HG-CRD) steps on a toy example when we start the diffusion process from node 0 and where each hyperedge (triangle in this case) has a maximum capacity of two.** Note that red nodes are nodes with excess of flow, while black nodes are nodes with no excess. At the end, HG-CRD was able to highlight the correct cluster containing nodes 0, 1, 2 and 3.

## 4.6 HG-CRD analysis

The analysis of CRD proceeds in a target recovery fashion. Suppose there exists a set $B$ with motif conductance $\phi$ and that is well-connected internally in a manner we will make precise shortly. This notion of internal connectivity is discussed at length in [30] and [33]. Suffice it to

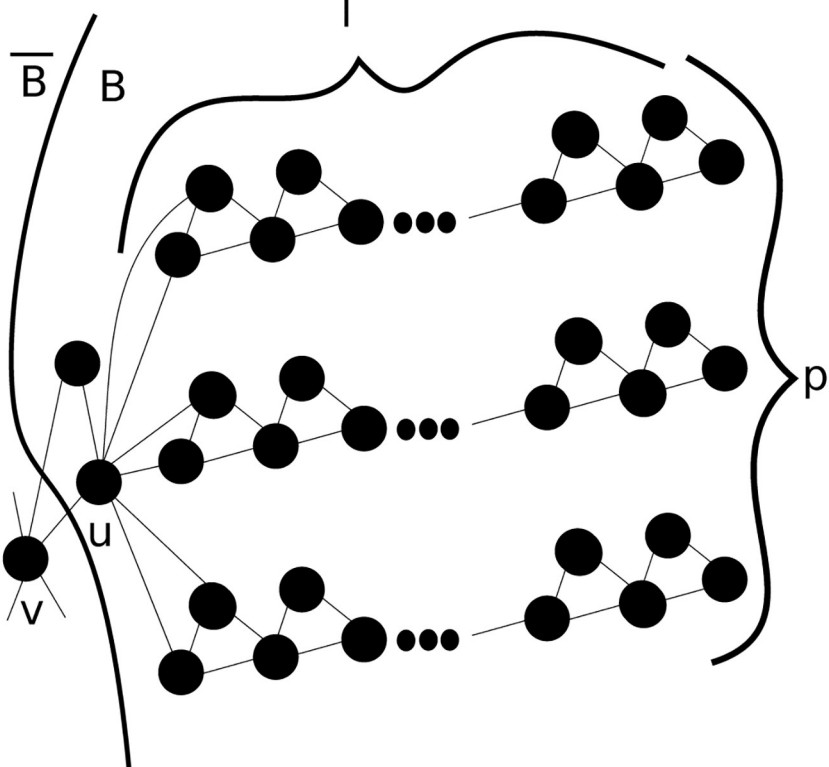

**Fig 3. A generalized example to show a comparison between hypergraph CRD (HG-CRD) and higher order spectral clustering where the triangles are the hyperedges.** Starting the diffusion process from node $u$, HG-CRD will stay more localized in cluster $B$ and will leak flow to $\bar{B}$ much less than higher order spectral clustering techniques by a factor of $\Omega\left(\frac{p}{\log l}\right)$.

say, the internal connectivity constraints that are likely to be true for much of the social and information networks studied, whereas these internal connectivity constraints are not likely to be true for planar or grid-like data. We show that HG-CRD, when seeded inside $B$ with appropriate parameters, will identify a set closely related to $B$ (in terms of precision and recall) with conductance at most $O(k\phi)$ where $k$ is the largest size of a hyperedge. Our theory heavily leverages the results from [30]. We restate theorems here for completeness and provide all the adjusted details of the proofs in the S1 File. However, these should be understood as mild generalizations of the results—hence, the statement of the theorems is extremely similar to [30]. Note that there are some issues with directly applying the proof techniques. For instance, some of the case analysis requires new details to handle scenarios that only arise with hyperedges. These scenarios are discussed in 4.6.1.

**Theorem 1 (Theorem 1 [30] Generalization).** *Given $G$, $\mathbf{m}_M$, $\phi \in (0, 1]$ such that: $\mathbf{e}^T\, \mathbf{m}_M \leq volM(G)$ and $\mathbf{m}_M(v) \leq 2\mathbf{d}_M(v)\ \forall v \in V$ at the start, HG-CRD inner terminates with one of the following:*

- ***Case 1**: HG-CRD-inner finishes with $\mathbf{m}_M(v) \leq \mathbf{d}_M(v)\ \forall v \in V$,*

- ***Case 2**: There are nodes with excess and we can find cut $A$ of motif-based conductance of $O(k\phi)$. Moreover, $2\mathbf{d}_M(v) \geq \mathbf{m}_M(v) \geq \mathbf{d}_M(v)\ \forall v \in A$ and $\mathbf{m}_M(v) \leq \mathbf{d}_M(v)\ \forall v \in \bar{A}$.*

Let us assume that there exists a good cluster $B$ that we are trying to recover. The goodness of cluster B will be captured by the following two assumptions, which are a generalized version of the two assumptions mentioned in CRD analysis:

**Assumption 1 (Assumption 1 [30] Generalization).**

$$\sigma_1 = \phi_M^{(s)}(B)/((k-1)\phi_M(B)) \geq \Omega(1),$$

*where $\phi_M^{(s)}(B)$ is the set motif conductance, which is defined as the minimum motif conductance of any set in the induced subgraph on B. This means that any subset in B has worse motif conductance than the set B by a gap of $k - 1$, which makes B a good cluster.*

**Assumption 2 (Assumption 2 [30] Generalization).**
$\exists\, \sigma_2 \geq \Omega(1)$, *such that any $T \subset B$ having $volM(T) \leq volM(B)/2$, satisfies:*

$$|M(T, B\ T)|/(|M(T, \bar{B})|\ log\ volM(B)/\phi_M^{(S)}(B)) \geq \sigma_2,$$

*where $|M(A, B)|$ is the number of hyperedges with at least one endpoint in A and at least another endpoint in B. This assumption states that any subset T in B is more connected via hyperedges to nodes in B than nodes in $\bar{B}$ by a factor of $\sigma_2\ log\ volM(B)/\phi_M^{(S)}(B)$.*

HG-CRD will work as follows, similar to CRD process, we will assume good cluster $B$ that satisfies assumption 1 and 2, and has the following properties: $\mathrm{vol}_M(B) \leq \mathrm{vol}_M(G)/2$, the diffusion will start from $v_s \in B$ and we know estimates of $\phi_M^{(S)}(B)$ and $\mathrm{vol}_M(B)$ and we will set $\phi = \theta(\phi_M^{(S)}(B)/k)$.

**Theorem 2 (Theorem 3 [30] Generalization).** *If we run HG-CRD with $\phi \geq \Omega(\phi_M(B))/k$, then:*

- $vol_M(A \setminus B) \leq O(k\,/\,\sigma).vol_M(B)$,

- $vol_M(B \setminus A). \leq O(k\,/\,\sigma)vol_M(B)$,

- $\phi_M(A) \leq O(k\phi)$,

*where $A = \{v \in V | d_M(v) \leq m_M(v)\}$ and $\sigma = min(\sigma_1, \sigma_2)$.*

**4.6.1 Theoretical challenges.** Proving the previous theorems for HG-CRD is non-trivial since several cases arises in hyperedges that do not exist in the original CRD. In this subsection, we discuss some of these non-trivial cases (The complete proofs of the theorems are provided in the S1 File), such as:

- Theorem 1: To prove this theorem, CRD [30] groups nodes based on their level value (nodes in level $i$ will be in group $B_i$) and then consider nodes with level at least $i$ to be in one cluster ($S_i$). After that, they categorize edges between cluster $S_i$ and $\bar{S}_i$ into two groups based on the level values of their endpoints. Since hyperedges do not have two endpoints, we needed to extend this categorization and proved similar properties for the extension. Because of this extension, we got a gap of $k$ between the motif conductance and $\phi$ as we proved that the motif conductance is $O(k\phi)$ instead of the original proof where the conductance is $O(\phi)$.

- Theorem 2: The proof of CRD [30] relies on the following equation:

$$\sum_{i=1}^{h} |E(B_i, \bar{B})| \min(i, 1/\phi) = \sum_{i=1}^{1/\phi} |E(S_i, \bar{B})|,$$

where $E(A, B)$ is the set of edges from cluster $A$ to cluster $B$. This equation only holds for edges, however this equation does not hold for hyperedges of $k > 2$. Therefore, we provide a totally different proof that works for hyperedges of any sizes.

## 4.7 Running time and space discussion

The running time of local algorithms are usually stated in terms of the output. Recall that CRD running time is $O((\text{vol}(A) \log \text{vol}(A))/\phi)$. As hypergraph CRD replaces the degree of vertices by the motif-based degree and the flow in each iteration depends on the motif-based degree, therefore the running time will depend on $\text{volM}(A)$ instead of $\text{vol}(A)$ and it will be $O((\text{volM}(A) \log \text{volM}(A))/\phi)$. For space complexity, it is $O(\text{volM}(A))$, as each node $v$ we explore in our local clustering will store hyperedges containing it.

## 4.8 HG-CRD extension for non-uniform hypergraphs

While the discussion so far is for uniform hypergraphs. HG-CRD can be easily extends to non-uniform hypergraphs (Hypergraphs containing hyperedges with different sizes). At the start, the flow value $\mathbf{m}_{M(s)}$ at the seed node $s$ will be equal to $2\mathbf{d}_M(s)$ and in each iteration, HG-CRD will double the flow on each visited node. Each node $v$ that has excess flow picks an *eligible* hyperedge $e$ that contains the node $v$ and sends flow to all other nodes in the hyperedge $e$. In this case, the eligibility definition of a hyperedge is extended to the following:

- $\mathbf{l}(v) > \mathbf{l}(u_i)$ for at least $\alpha$ $u_i \in e \setminus v$, where $\alpha \in [1, k-1]$.

- $r(e) > 0$.

- $ex(v) \geq |e| - 1$

After performing HG-CRD-inner iteration, we will remove any excess flow at the nodes. If all nodes do not have excess flow in all iterations (no bottleneck is reached yet) and as we double the flow at each iteration $j$, then the total sum of flow at the nodes will be equal to $2\mathbf{d}_M(s)2^j$. However, if the total sum of flow at the nodes is significantly (according to the parameter $\tau$) less than $2\mathbf{d}_M(s)2^j$, then step 8 has removed a lot of flow excess at the nodes because many nodes had flow excess at the end of the iteration. This indicates that the flow was contaminated and we have reached a bottleneck for the cluster.

Additionally, we evaluate HG-CRD on a non-uniform hypergraph called mathoverflow-answers in the experiments section.

## 5 Experimental results

In this section, we will compare CRD using motif adjacency matrix (CRD-M) and hypergraph CRD (HG-CRD) to the original CRD in the community detection task using both synthetic datasets and real world graphs, then we will compare HG-CRD to other related work like motif-based approximate personalized PageRank (MAPPR) and approximate personalized PageRank (APPR) in the community detection task using both undirected and directed graphs. Table 2 shows the characteristics of the datasets used in the experiments. You can download SNAP datasets from https://snap.stanford.edu/data/ and large hypergraphs from https://www.cs.cornell.edu/~arb/data/. All experiments were done using Python 3.6.4 or Julia 1.4 on Windows environment. The experiments run on a single computer with 8GB RAM, Intel core i3 processor and 1TB hard drive.

Additionally (Section 5.6), we evaluation hypergraph CRD (HG-CRD) on a number of hypergraphs that have recently been analyzed. These are networks where each hyperedge here involves far more than one node, with average hyperedge sizes around 20, and there are fewer hyperedges than total nodes. The ones we use were recently studied from a different perspective on hypergraph clustering using flow-based techniques [34]. The hypergraphs can be downloaded from https://www.cs.cornell.edu/~arb/data/. We compare against this recent code and the final experiment does not show a conclusive result as far as which is uniformly better, but does identify a case where HG-CRD does outperform the recently proposed hyperlocal flow-based framework.

**Reproduction**: You can find the source code to regenerate all the figures and the tables in the .zip folder in https://www.dropbox.com/s/p3ap5j7ld4adphb/Higher_order_capacity_releasing_diffusion-master.zip?dl=0.

### 5.1 Synthetic dataset

We use the LFR model [35] to generate the synthetic datasets as it is a widely used model in evaluating community detection algorithms. The parameters used for the model are $n = 1000$, average degree is 10, maximum degree is 50, minimum community size is 20 and maximum community size is 100. LFR node degrees and community sizes are distributed according to the power law distribution, we set the exponent for the degree sequence to 2 and the exponent for the community size distribution to 1. We vary $\mu$ the mixing parameter from 0.02 to 0.5 with step 0.02 and run CRD, CRD using motif adjacency matrix (CRD-M) and hypergraph CRD (HG-CRD) using a triangle as our desired higher order pattern. Each technique is run 100 times from random seed nodes and report the median of the results. The implementation of CRD requires four parameters which are maximum capacity per edge $C$, maximum level of a node $h$, maximum iterations $t$ of CRD inner, how much excess of flow is too much $\tau$. We use

**Table 2. Characteristics of datasets used in the experiments, where |V| is the number of nodes, |E| is the number of edges, |C| is the number of communities and sizes is the range of the number of nodes in each community.**

| Dataset | |V| | |E| | |C| (sizes) |
|---------|-----|-----|-------------|
| DBLP | 317,080 | 1,049,866 | 100 (10–36) |
| Amazon | 334,863 | 925,872 | 100 (10–178) |
| YouTube | 1,134,890 | 2,987,624 | 100 (10–200) |
| Email-EU | 1,000 | 25,600 | 28 (10–109) |

Note that the number of communities and their sizes are chosen as the same as MAPPR [16] and CRD [30].

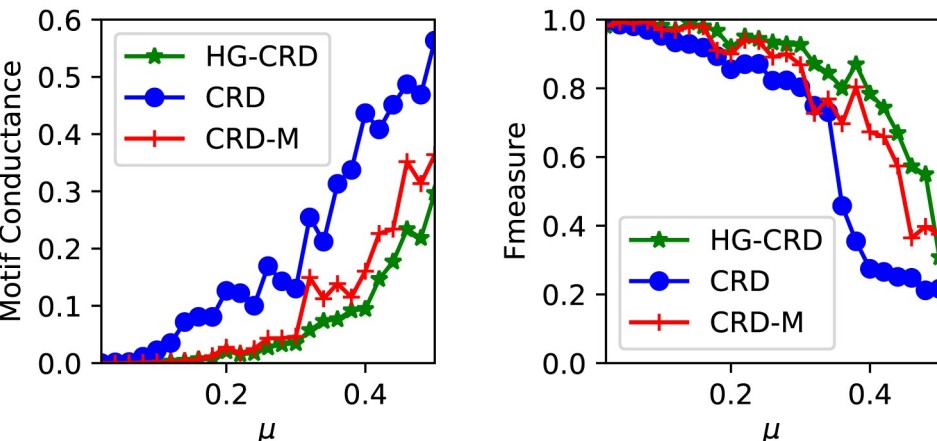

**Fig 4. Comparison between CRD, CRD using motif adjacency matrix (CRD-M) and hypergraph CRD (HG-CRD) on LFR synthetic datasets.** We don't discuss some of the PageRank algorithms as those are outperformed by CRD as shown in previous experiments [30].

the same parameters for all CRD variations following the setting in [30], $h = 3$, $C = 3$, $\tau = 2$ and $t = 20$. For HG-CRD, we set $\alpha$ to be 1. As shown in Fig 4, HG-CRD has the lowest motif conductance and it has better $F_1$ than the original CRD and CRD-M. HG-CRD gets higher $F_1$ when communities are harder to recover, when $\mu$ gets large.

## 5.2 Hypergraph-CRD compared to CRD

Local community detection is the task of finding the community when given a member of that community. In this task, we start the diffusion from the given node. Table 3 shows the community detection results for CRD, CRD using motif adjacency matrix (CRD-M) and hypergraph CRD (HG-CRD). In these experiments, we follow the experiment setup in [16], we identify 100 communities from the ground truth such that each community has a size in the range mentioned in Table 2. The community sizes are chosen similar to the ones reported in [16] and [30]. Then for all algorithms, following the evaluation setup used in [16], we start from each node in the community and then report the result from the node that yields the best F1 measure. Finally, we average the F1 scores over all the detected communities for the method. The implementation of CRD requires four parameters which are maximum capacity per edge $C$, maximum level of a node $h$, the maximum number of times $t$ that CRD inner is called, how much excess of flow is too much $\tau$. We use the same parameters for all CRD variations following the setting in [30], which are: $C = 3$, $h = 3$, $t = 20$ and $\tau = 2$ except in Youtube, we set the maximum number of iterations $t$ to be 5 as the returned community size was very large and therefore the precision was small (Increasing the maximum number of iterations increases the returned community size as it allows the algorithm to explore wider regions). Additionally, we

**Table 3. Comparison between CRD, CRD-M and HG-CRD using SNAP datasets.**

|  | Motif Conductance | | | | $F_1$ | | | |
|---|---|---|---|---|---|---|---|---|
|  | CRD | CRDM | HGCRD $\alpha = 1$ | HGCRD $\alpha = 2$ | CRD | CRDM | HGCRD $\alpha = 1$ | HGCRD $\alpha = 2$ |
| DBLP | 0.489 | 0.471 | **0.394** | 0.451 | 0.313 | 0.320 | 0.329 | **0.359** |
| Amazon | 0.267 | 0.159 | **0.122** | 0.194 | **0.632** | 0.603 | 0.585 | 0.561 |
| YouTube | 0.892 | 0.807 | **0.772** | 0.785 | 0.162 | 0.165 | 0.171 | **0.175** |

choose the triangle as our specified motif and for HG-CRD, we try all possible values of $\alpha$, which are 1 and 2 and report the results of both versions. As shown in Table 3, hypergraph CRD (HG-CRD) has a lower motif conductance than CRD in all datasets and it has the best or close $F_1$ in all datasets except Amazon. Looking closely, we can see that hypergraph CRD has a higher precision than the original CRD in all datasets. This can be attributed to exploiting the use of motifs which kept the diffusion more localized in the community. Additionally, CRD using motif adjacency matrix (CRD-M) has lower motif conductance than CRD in four datasets and higher $F_1$ in three datasets. The higher $F_1$ of CRD-M can also be attributed to the higher precision it achieves over CRD.

### 5.3 Related work comparison

In this section, we will compare hypergraph CRD to motif-based approximate personalized PageRank (MAPPR) and approximate personalized PageRank (APPR) in the community detection task. We follow the same experimental setup as mention in the previous section. Table 4 shows the precision, recall and $F_1$ for hypergraph CRD, MAPPR and APPR. As shown in Table 4, HG-CRD obtains the best $F_1$ in all datasets and a higher precision than MAPPR in all datasets by up to around 10% in the YouTube dataset. This enhancement can be attributed to the CRD dynamics where it keeps the diffusion localized while spectral techniques yield more leakage outside the community.

Furthermore, we compare HG-CRD to CRD, APPR and MAPPR using a directed graph, which is Email-EU. We set the parameters to be the same as the last section, and set $\alpha$ to be 1. For this task, we try three different directed motifs shown in Fig 5, which are a triangle in any direction (M1), a cycle (M2), and a feed-forward loop (M3). As shown in Table 5 HG-CRD has the highest $F_1$ by around 10% compared to MAPPR, which again attributed to its high precision.

### 5.4 Running time experiments

We have made no extreme efforts to optimize running time. Nevertheless, we compare the running time of CRD, CRD-M and HG-CRD on LFR datasets while varying the mixing

**Table 4. Comparison between HG-CRD and MAPPR using undirected and directed graphs.**

| | Precision | | | Recall | | | $F_1$ | | |
|---|---|---|---|---|---|---|---|---|---|
| | APPR | MAPPR | HGCRD | APPR | MAPPR | HGCRD | APPR | MAPPR | HGCRD |
| DBLP | 0.342 | 0.366 | **0.404** | 0.310 | 0.329 | **0.362** | 0.264 | 0.269 | **0.329** |
| Amazon | 0.634 | 0.660 | **0.718** | **0.704** | 0.567 | 0.566 | **0.620** | 0.567 | 0.585 |
| YouTube | 0.233 | 0.390 | **0.485** | 0.147 | **0.188** | 0.162 | 0.140 | 0.165 | **0.172** |

The numbers for APPR and MAPRR are taken from [16].

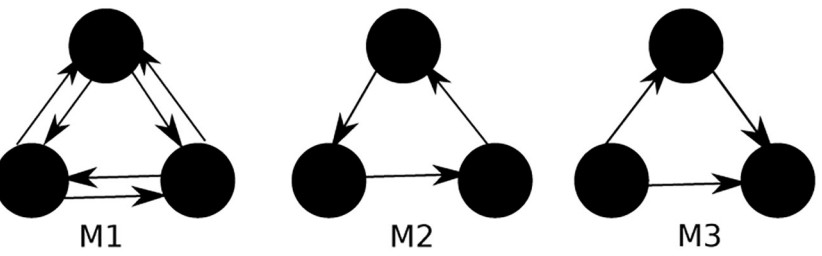

**Fig 5. Three directed motifs used for clustering Email-EU graph, which are a triangle in any direction (M1), a cycle (M2), and a feed-forward loop (M3).**

**Table 5. Comparison between higher order CRD (HG-CRD) and motif-based approximate personalized pageRank (MAPPR) on directed Email-EU graph.**

|  | Precision | Recall | $F_1$ |
|---|---|---|---|
| APPR | 0.502 | **0.754** | 0.398 |
| CRD | 0.801 | 0.394 | 0.496 |
| MAPPR using M1 | 0.580 | 0.685 | 0.496 |
| MAPPR using M2 | 0.605 | 0.577 | 0.443 |
| MAPPR using M3 | 0.660 | 0.594 | 0.483 |
| CRD-M using M1 | 0.793 | 0.492 | 0.607 |
| CRD-M using M2 | 0.715 | 0.467 | 0.521 |
| CRD-M using M3 | 0.793 | 0.492 | 0.607 |
| HG-CRD using M1 | **0.808** | 0.504 | **0.621** |
| HG-CRD using M2 | 0.777 | 0.515 | 0.587 |
| HG-CRD using M3 | **0.808** | 0.504 | **0.621** |

The numbers for APPR and MAPRR are taken from [16].

parameter ($\mu$). Fig 6 shows the running times of detecting a community of a single node, we repeat the run 100 times starting from random nodes, then we report the mean running time and the error bars represent the standard deviations. As shown in the figure, when the communities are well separated ($\mu$ is less than 0.3), CRD is the fastest technique. HG-CRD is slower than CRD by a small gap 0.1 seconds and is faster than CRD-M. When the communities are hard to recover ($\mu$ gets larger than 0.3), CRD takes a longer time to recover the communities. However, both HG-CRD and CRD-M are able to recover the communities faster and with higher quality since they use higher order patterns.

## 5.5 Robustness experiments

We have filtered both com-DBLP and com-Amazon datasets to only include clusters with conductance less than 0.5 and volume at least 1000. We run each method starting from each node

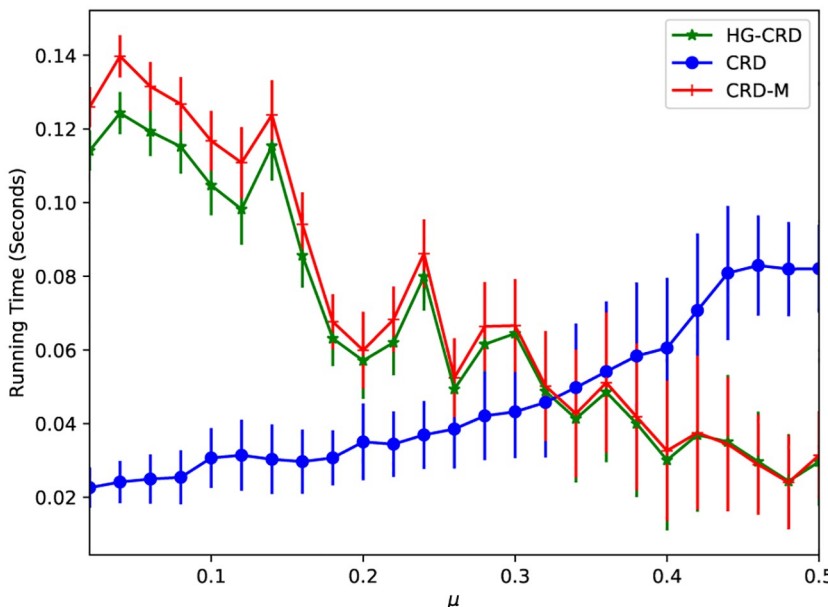

**Fig 6. Running time comparision between CRD, CRD-M and HG-CRD on LFR datasets.**

in the cluster and then report the median motif conductance and the median F1 along with the distribution over the choices of seed to show the robustness of the techniques.

As shown in S1 and S2 Tables, HG-CRD always enhances the F1 measure in com-DBLP dataset for all communities. It also maintains a comparable motif conductance to other techniques except in DBLP-595 where it has much better F1 measure. For com-Amazon dataset, all techniques had comparable F1 and motif conductance in all communities.

## 5.6 Large hyperedge experiments

We finally mirror an experiment from a recent hypergraph clustering paper [34], where the Hyperlocal algorithm and some alternative baseline measures originate. These are all designed for large hyperedges.

We use two question-user datasets, one based on Math Overflow and the second one on Stack Overflow. Each node represents a question and each hyperedge represents a set of questions answered by users. As such, these networks have a small number of hyperedges that cover the nodes. Hyperedges have large average sizes. In Math Overflow, there are 73,851 nodes and 5,446 hyperedges. Each question is labeled by a set of tags and questions often have multiple labels. The mean size of a hyperedge is 24.2 and the maximum hyperedge size is 1,784. This dataset contain 1,456 class given by the tags on questions. Our goal is to recover the tags starting from a seed. The Stack Overflow experiment is slightly different and we describe it below.

These techniques all assume knowledge of the true cluster size whereas our Hypergraph CRD algorithm does not.

- **TopNeighbors (TN)**: This baseline orders nodes in the one hop neighborhood of the seed node based on the number of hyperedges that each neighbor shares with the seed node and outputs the top $k$ such nodes. We set $k$ equals to the ground truth class size.

- **BestNeighbors (BN)**: This baseline orders each node $v$ in the one hop neighborhood of the seed node by the fraction of hyperedges containing $v$ that also contains the seed node and finally outputs the top $k$. We set $k$ equals to the ground truth class size.

- **Hyperlocal**: This technique is introduced in [34]. Its objective is to refine an existing cluster based on hypergraph conductance based objective function. Similar to the evaluation of the paper, we input the BN clusters as the existing clusters for Hyperlocal to refine.

We start all the algorithms from one random seed node and repeat the experiment 5 times for each cluster. For HG-CRD, we set $C = 3$, $h = 3$, $t = 3$ and $\tau = 2$ and for Hyperlocal, we set $\varepsilon = 1.0$ and use the $\delta$-linear threshold penalty. We have tried both $\delta = 1.0$ and $\delta = 5000.0$ and report the one with the best F1-measure (here that was $\delta = 1$ for the Math Overflow) which was how the parameters were used in Hyperlocal paper [34]. We could not run CRD-M since this will require replacing each hyperedge with a clique expansion and then running CRD-M on the expanded graph. This results in a very large graph as in Math Overflow dataset, we are replacing 5,446 hyperedges that on average have 24.2 nodes; in the future we may explore lazy expansion versions to further optimize this technique.

We report the average precision, recall, F1-measure across all the 1,456 clusters of the dataset in Table 6. As shown in the table, HGCRD has the best F1-measure and this is attributed to having much higher precision than the other techniques. This is again attributed to having higher precision than other techniques. Also, note that HGCRD method does not utilize knowledge of the true cluster size, whereas other techniques do.

**Table 6. Precision, recall and F1-measure for TopNeighbors (TN), BestNeighbors (BN), Hyperlocal (HL) and HG-CRD on mathoverflow-answers hypergraph and five randomly chosen classes from stackoverflow-answers hypergraph dataset (which are relative-time-span, type-conversion, binary-data, zos and mainframe).**

|  | Precision | | | | Recall | | | | F1-measure | | | |
|---|---|---|---|---|---|---|---|---|---|---|---|---|
|  | TN | BN | HL | HGCRD | TN | BN | HL | HGCRD | TN | BN | HL | HGCRD |
| math-overflow | 0.30 | 0.30 | 0.30 | **0.78** | **0.46** | 0.46 | 0.30 | 0.44 | 0.36 | 0.36 | 0.36 | **0.49** |
| relative-time-span | 0.10 | 0.10 | 0.10 | **0.80** | 0.11 | 0.11 | 0.11 | 0.11 | 0.11 | 0.11 | 0.11 | **0.19** |
| type-conversion | 0.05 | 0.05 | 0.59 | **0.69** | 0.05 | 0.05 | 0.05 | 0.05 | 0.05 | 0.05 | **0.09** | **0.09** |
| binary-data | 0.05 | 0.05 | 0.46 | **0.80** | 0.05 | 0.05 | 0.05 | 0.05 | 0.05 | 0.05 | 0.08 | **0.09** |
| zos | 0.13 | 0.14 | 0.50 | **0.64** | 0.13 | **0.15** | 0.08 | 0.05 | 0.13 | **0.14** | 0.13 | 0.10 |
| mainframe | 0.31 | 0.23 | 0.49 | **0.77** | **0.32** | 0.24 | 0.15 | 0.06 | **0.32** | 0.23 | 0.23 | 0.11 |

In the Stack Overflow experiment, we have a bigger hypergraph with 15,211,989 nodes and 1,103,243 hyperedges. The mean size of a hyperedge is 23.7 and the maximum hyperedge size is 61,315 and the dataset contain 56,502 class. This experiment differs in the seeding and we randomly select 5% from each ground truth class and use them as the seed nodes. The parameters of HGCRD are the same. For Hyperlocal, we again tried both $\delta = 1$ and $\delta = 5000$ and this experiment had the best F1 results using $\delta = 5000$ (compared with $\delta = 1$ Math Overflow). Table 6 shows the F1-measure for five randomly selected classes (relative-time-span, type-conversion, binary-data, zos and mainframe). For each class, we repeat the experiment 5 times. As shown in the table, there are some classes where HGCRD obtains better results. Again, let us point out that this does not assume knowledge of the true cluster sizes.

## Supporting information

**S1 Table. Comparison between CRD, CRD-M, HG-CRD, APPR and MAPPR using SNAP datasets.** Each column represents a community in a SNAP dataset, for example DBLP-104 means the community number 104 in com-DBLP dataset. We report the median of F1 and the violin plots show the distribution while varying the seed node.
(PDF)

**S2 Table. Comparison between CRD, CRD-M, HG-CRD, APPR and MAPPR using SNAP datasets.** Each column represents a community in a SNAP dataset, for example DBLP-104 means the community number 104 in com-DBLP dataset. We report the median of motif conductance and the violin plots show the distribution while varying the seed node.
(PDF)

**S1 File.**
(PDF)

## Acknowledgments

Both authors would like to acknowledge our colleagues who provided initial feedback on this paper.

## Author Contributions

**Conceptualization:** Rania Ibrahim.

**Formal analysis:** Rania Ibrahim.

**Investigation:** Rania Ibrahim, David F. Gleich.

**Methodology:** Rania Ibrahim.

**Software:** Rania Ibrahim.

**Validation:** Rania Ibrahim.

**Writing – original draft:** Rania Ibrahim.

**Writing – review & editing:** Rania Ibrahim, David F. Gleich.

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
