## [Decision Letter · Decision Letter 0]

20 Oct 2020

PONE-D-20-27457

Local hypergraph clustering using capacity releasing diffusion

PLOS ONE

Dear Dr. Ibrahim,

Thank you for submitting your manuscript to PLOS ONE. After careful consideration, we feel that it has merit but does not fully meet PLOS ONE’s publication criteria as it currently stands. Therefore, we invite you to submit a revised version of the manuscript that addresses the points raised during the review process.

From the reports received, one of the reviewers is raising some concerns with the experiments performed and the methodology used to withdraw some of the conclusions. They also pointed out some minor typos to amend. 

We look forward to receiving your revised manuscript.

Kind regards,

Irene Sendiña-Nadal

Academic Editor

PLOS ONE

Journal Requirements:

2) We note you have included a table to which you do not refer in the text of your manuscript. Please ensure that you refer to Table 1 in your text; if accepted, production will need this reference to link the reader to the Table.

Reviewers' comments:

Reviewer's Responses to Questions

**Comments to the Author**

1. Is the manuscript technically sound, and do the data support the conclusions?

Reviewer #1: Yes

Reviewer #2: Partly

2. Has the statistical analysis been performed appropriately and rigorously? 

Reviewer #1: Yes

Reviewer #2: N/A

3. Have the authors made all data underlying the findings in their manuscript fully available?

Reviewer #1: Yes

Reviewer #2: Yes

4. Is the manuscript presented in an intelligible fashion and written in standard English?

Reviewer #1: Yes

Reviewer #2: Yes

5. Review Comments to the Author

Reviewer #1: In this manuscript the authors place their research within the context of existing literature and extend the recent technique called CDR (capacity releasing diffusion, a method to find local clusters presented in 2017)

to a new method called hypergraph CRD (HG-CRD).

This new method outperforms the capabilities of the forementioned CDR in the sense that it provides a more robust computation of the conductance associated to the used motifs. The new method

is important for two reasons. On the one hand, this new framework

makes use of an hypergraph (or multigraph) defined on the same nodes that the original graph in which is wanted to perform

the clustering, jointly with a set of hyperedges (sets of nodes connected by a pattern or motif). On the other hand, HG-CRD does not need the construction of the motif-weighted adjacency matrix (and this is a difference with some methods for local clustering based on Personalized PageRank). The authors also extend some known theoretical results to the new framework. One of the great capabilities of the presented method is the ability to use, in principle, any kind of motif, including directed subgraphs. The algorithm is not subjected to the

Cheeger inequality when there is a proper cluster close to the seed nodes where the algorithm starts.

The results are obtained by using synthetic and real graphs and show that HG-CRD properly finds motif clusters and it may

be a useful tool for community detection in a wide range of applications. (See the attached file)

Reviewer #2: See the attached pdf.

6. PLOS authors have the option to publish the peer review history of their article (what does this mean?). If published, this will include your full peer review and any attached files.

Reviewer #1: **Yes: **Francisco Pedroche

Reviewer #2: No

---

## [Author Response · Author response to Decision Letter 0]

5 Nov 2020

We would like to thank the reviewers for the comments and time to improve our paper. Please find our reply for each point raised by the reviewers. Also, see the attached \\emph{diff} that shows how these changes have impacted the manuscript. 

The largest change is the addition of a new \\emph{robustness} section in response to Reviewer Two's comments on the performance of the method. This shows numerous scenarios where our method does quite well. 

Kindly find the updated version, our reply and the file with marked changed.

---

## [Decision Letter · Decision Letter 1]

23 Nov 2020

Local hypergraph clustering using capacity releasing diffusion

PONE-D-20-27457R1

Dear Dr. Ibrahim,

We’re pleased to inform you that your manuscript has been judged scientifically suitable for publication and will be formally accepted for publication once it meets all outstanding technical requirements.

Kind regards,

Irene Sendiña-Nadal

Academic Editor

PLOS ONE

Additional Editor Comments (optional):

Reviewers' comments:

Reviewer's Responses to Questions

**Comments to the Author**

1. If the authors have adequately addressed your comments raised in a previous round of review and you feel that this manuscript is now acceptable for publication, you may indicate that here to bypass the “Comments to the Author” section, enter your conflict of interest statement in the “Confidential to Editor” section, and submit your "Accept" recommendation.

Reviewer #1: All comments have been addressed

Reviewer #2: (No Response)

2. Is the manuscript technically sound, and do the data support the conclusions?

Reviewer #1: Yes

Reviewer #2: Yes

3. Has the statistical analysis been performed appropriately and rigorously? 

Reviewer #1: Yes

Reviewer #2: Yes

4. Have the authors made all data underlying the findings in their manuscript fully available?

Reviewer #1: Yes

Reviewer #2: Yes

5. Is the manuscript presented in an intelligible fashion and written in standard English?

Reviewer #1: Yes

Reviewer #2: Yes

6. Review Comments to the Author

Reviewer #1: The manuscript has been modified according to some suggestions of the referees. Besides correcting some minor typos about language and structrure, some new explanations and experiments have been added. In my view, the manuscript is now more readable and it can be published in the current form.

Reviewer #2: The authors replied to all my concerns sufficiently and they did all the additional necessary experiments. I recommend accepting the paper.

7. PLOS authors have the option to publish the peer review history of their article (what does this mean?). If published, this will include your full peer review and any attached files.

Reviewer #1: **Yes: **Francisco Pedroche

Reviewer #2: No

---

## [Editor Report · Acceptance letter]

25 Nov 2020

PONE-D-20-27457R1 

Local hypergraph clustering using capacity releasing diffusion 

Dear Dr. Ibrahim:

I'm pleased to inform you that your manuscript has been deemed suitable for publication in PLOS ONE. Congratulations! Your manuscript is now with our production department. 

Kind regards, 

on behalf of

Dr. Irene Sendiña-Nadal 

Academic Editor

PLOS ONE